# ADAPTIVE REPRESENTATION SELECTION IN CONTEXTUAL BANDIT WITH UNLABELED HISTORY

**Baihan Lin**
Department of Neuroscience and Statistics, Columbia University
`Baihan.Lin@columbia.edu`

**Guillermo A. Cecchi, Djallel Bouneffouf, & Irina Rish**
IBM Thomas J. Watson Research Center
`{gcecchi, dbouneffouf, rish}@us.ibm.com`

## ABSTRACT

We consider an extension of the contextual bandit setting, motivated by several practical applications, where an unlabeled history of contexts can become available for pre-training before the online decision-making begins. We propose an approach for improving the performance of contextual bandit in such setting, via adaptive, dynamic representation learning, which combines offline pre-training on unlabeled history of contexts with online selection and modification of embedding functions. Our experiments on a variety of datasets and in different nonstationary environments demonstrate clear advantages of our approach over the standard contextual bandit.

## 1 INTRODUCTION

In the *contextual multi-armed bandit (CMAB)*, at each iteration, before choosing an arm, the agent observes an $N$-dimensional *context*, or *feature vector*. Over time, the goal is to learn the relationship between the context vectors and the rewards, in order to make better prediction which action to choose given the context (Agrawal & Goyal, 2013; Langford & Zhang, 2008a). However, in certain real-life applications, before the online decision-making starts, an agent may have an access to a *unlabeled context history* (i.e., contexts without the associated rewards). For instance, in medical decision-making settings (Villar et al., 2015), the doctor may have an access to medical records of different patients, which can be used to gain a better understanding of the patients population.

Having an access to unlabeled data makes it possible to *pre-train some model of the input in an offline mode*, and use it later to improve the online decision making. For example, we can learn an autoencoder to map the raw inputs into potentially better representations. Moreover, when the inputs are non-homogeneous, we may want to cluster the unlabeled data and learn separate representations for each cluster. Then, in the online mode, we can decide which representation to use for a given context; such *context-driven representation selection* has a potential to further improve the subsequent decision-making. These representation models can continue to be updated online as more contexts become available, especially in *nonstationary environments* abundant in practical applications, where both the context distribution and the reward distribution can change.

Motivated by the above scenarios, we consider here a contextual bandit setting, called *Contextual Bandit with Representation learning and unlabeled History (CBRH)*. In this setting, it is assumed that (1) *a set of unlabeled contexts is available for pre-training* before the online decision-making starts, (2) the bandit's performance can be improved by *learning a good context representation (embedding)* rather than using the raw input, the (3) embedding functions are *pre-trained on the unlabeled history* and *adaptively selected (and updated) based on the context* during the online decision-making. Next, we propose an algorithm for the above CBRH setting, called *Adaptive Bandit with Context-Driven Embeddings (ABaCoDE)*, which implements online, clustering-based embedding selection and learning coupled with Thompson-Sampling contextual bandit approach.

## 2 ADAPTIVE BANDIT WITH CONTEXT-DEPENDENT EMBEDDINGS (ABaCoDE)

We now describe an adaptive, context-driven embedding selection approach to solving the CBRU problem. It has two variants, based on online- and offline clustering, respectively; the choice is controlled by a Boolean input parameter $isOnline$ in Algorithm 1. Two more inputs include: an unlabeled pre-training dataset $\mathbf{D}$, as well as the number of embeddings $k$. The algorithm processes the input contexts sequentially, one by one, but at the end of each *mini-batch* of data it updates the embeddings to reflect possible changes in the data distribution. The initialization step (line 2) con-

---

**Algorithm 1** Adaptive Bandit with Context-Dependent Embeddings (ABaCoDE)

---

1: **Input:** unlabeled dataset $\mathbf{D}$, a set of unlabeled contexts for pre-training; $k$, the number of clusters (and corresponding embeddings); a Boolean variable $isOnline$.
2: **Initialization:**     Cluster $\mathbf{D}$ into $k$ clusters: $\mathbf{C} = \{c_1, ..., c_k\}$,     For each cluster, train an autoencoder to construct a set of encoding functions (embeddings): $\mathbf{E} = e_1, ..., e_k$,     Initialize the contextual Thompson Sampling parameters of bandit $B$ (line 1 in Alg. 1).
3: **while** there is a next data mini-batch $\mathbf{M}$, **do**
4:     **foreach** $x_t$ from $\mathbf{M}$ **do**
5:         $c_j = assignCluster(x_t)$
6:         **if** $isOnline$ **then** $updateCluster(\mathbf{C}, x_t, c_j)$
7:         $e = selectEmbedding(c_j)$
8:         $z = e(x_t)$ (encoded context/representation)
9:         $contextualBandit(B, z)$
        **end**
10:     **if** $not(isOnline)$ **then** $recomputeClusters(\mathbf{C}, \mathbf{B})$
11:     $updateEmbedding(\mathbf{M}, \mathbf{C})$
    **end**

---

sists of clustering the pre-training dataset $\mathbf{D}$ into $k$ clusters, training an autoencoder for each cluster, which results into $k$ encoding (embedding) functions, and initializing parameters of the contextual Thompson Sampling bandit, used later to make classification decisions based on embedded context.

Next, the algorithm switches to the online mode, processing an online stream of incoming samples (contexts). As mentioned above, we assume that at the end of each fixed-length time window, i.e. a fixed-size mini-batch of data, we update our embeddings.

Within each data mini-batch $\mathbf{M}$ (line 4), once the next input sample $\mathbf{x}_i$ arrives, it is first assigned to one of the existing clusters $c_j$ (line 5), associated with the corresponding embedding function $e_j$. Next, an online clustering is performed if $isOnline$ is true, i.e. the centroid of the cluster $c_j$ is recomputed, but no changes are made to other clusters (line 6). Otherwise, there are no changes to clusters, until the end of the batch, as we will see shortly. Based on the cluster assignment $c_j$, the corresponding embedding function $e_j$ is used to compute the representation vector $z$ for given input $\mathbf{x}_i$ (line 7); given the context $z$, the contextual bandit $B$ makes a decision (line 8), obtains the reward $r_i$ (line 9), and updates its parameters (line 10) using the contextual Thompson Sampling described in the previous section. After the end of the mini-batch $\mathbf{M}$ is reached (line 11), if $isOnline$ was false, the clusters will be recomputed from scratch using all data points received so far (however, no such re-clustering is performed if the online clustering was selected). Finally, the embeddings are updated respectively using the updated set of clusters $\mathbf{C}$.

## 3 EMPIRICAL EVALUATION

We present empirical results comparing both online and offline clustering methods outlined above with two baseline approaches: *Contextual Bandit (CB)*: as the baseline, we use the standard contextual multi-armed bandit with Thompson Sampling, based on the raw input (i.e., no embeddings). *universal embedding (uE)*: a universal embedding denotes a single embedding computed based on all data, and always recomputed to include the data from the most recent mini-batch; no clustering is performed. *mini-batch embedding (mE)*: this is our offline clustering approach presented in Algo-

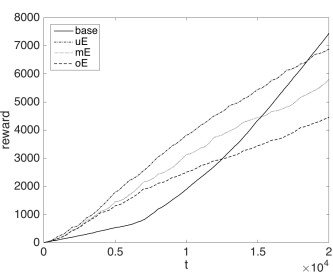
**Figure 1:** MNIST unshuffled, k = 2

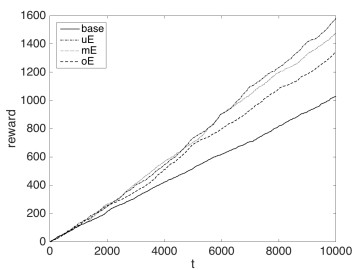
**Figure 2:** STL-10 unshuffled, k = 2

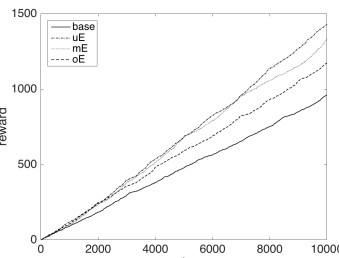
**Figure 3:** CIFAR-10 unshuffled, k = 2

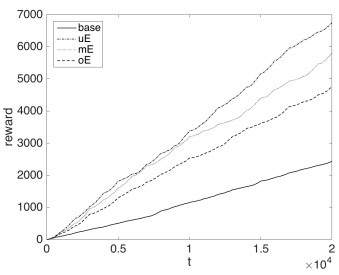
**Figure 4:** MNIST shuffled, k = 2

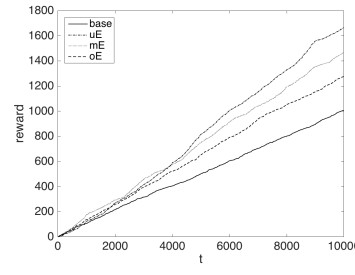
**Figure 5:** STL-10 shuffled, k = 2

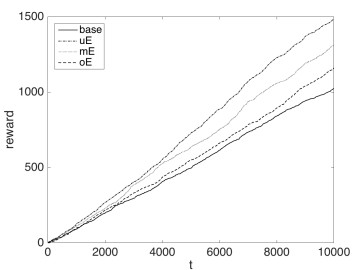
**Figure 6:** CIFAR-10 shuffled, k = 2

rithm 1, when $isOnline$ is $false$. *online embedding (oE)*: this is the online version of our algorithm described above, i.e. $isOnline$ is $true$.

We studied several types of nonstationarity by varying cluster distribution, or by introducing negative images as inputs with same semantics but different textures. Another type of nonstationarity was assuming that input samples may come from different tasks, and thus can be associated with different subsets of arms. we have also explored the multi-task setting by introducing a different type of nonstationary reward, where the class labels were shuffled, i.e. randomly permuted, in each batch.

For example, for MNIST dataset (Figure 1), we observe that initially, the baseline CB (solid line) is considerably worse than embedding-based approaches, and requires a large number of iteration to finally catch up with them. Figures 2 and 3 show the history of reward accumulation for the STL-10 and CIFAR-10, demonstrating that the baseline is consistently outperformed by embedding selection methods.

*Nonstationary due to varying cluster distribution, and for multi-task settings:* Our embedding-based approaches always outperformed the baseline, suggesting that in a setting where reward functions are nonstationary, in addition to the nonstationary input environment, the advantage of representation learning is quite significant, as compared to standard CB. Note that, with nonstationary (shuffled) labels, the reward accumulated by the baseline CB remains significantly below the reward of embedding-based approaches, at all iterations (Figures 4-6). Thus, in a more challenging setting with both context and reward nonstationarities, the embedding-based approaches clearly outperform the standard contextual bandit.

*Nonstationary setting with negative environments and unshuffled reward:* Again, the embedding-based approaches are always superior to the baseline CB; *online embedding* achieved the best performance among all methods on MNIST, while universal and batch embeddings were taking their turns outperforming the baseline on other datasets and settings.

*Nonstationary setting with negative environments and shuffled reward function:* the difference of textures under the same semantics introduced in this experiments demonstrated that embedding selection outperforms single universal embedding in most nonstationary cases.

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

# A    PROBLEM FORMULATION

Using the notation introduced in the previous section, we now define our novel bandit setting: *Contextual Bandit with Representation learning and unlabeled History (CBRH)* (outlined in Alg. 2), based on the following key assumptions.

First, we assume that *a context* $\mathbf{x}_t \in \mathbf{R}^N$ *is mapped into its representation* $\mathbf{z}_t \in \mathbf{R}^{N_i}$ using an *embedding* function $e_i(\mathbf{c}_t)$, selected from a set $E = \{e_1, ..., e_k\}$ of currently available embedding functions. Second, we assume that *the set of embedding functions E can be modified online*. And third, an access to *a set* $\mathbf{D}$ *of unlabeled contexts*, i.e. contexts without the associated rewards, is assumed. This dataset can be used, for example, for *pre-training* embedding functions $e(\mathbf{x})$. We then define a set $\Pi = \cup_{e_i \in E}\{\pi : \mathbf{R}^N \to A, \ \pi(\mathbf{c}) = \hat{\pi}(e_i(\mathbf{c}))\}$ of compound-function policies, where the function $\hat{\pi} : \mathbf{R}^{N_i} \to A$ maps $\mathbf{z}_t = e_i(\mathbf{c_t})$ to an action in $A$. The objective is to learn a hypothesis $\pi$ over $T$ iterations maximizing the cumulative reward.

---

**Algorithm 2** The CBRH Problem Setting

---

 1: Obtain unlabeled set of contexts $\mathbf{D}$
 2: Learn a context representation model
 3: **Repeat**
 4:     $(\mathbf{x}_t, r_t)$ is drawn according to distribution $D_{c,r}$
 5:     A context representation $\mathbf{z}_t$ is obtained
 6:     The player chooses an arm $k_t = \hat{\pi}(\mathbf{z}_t)$
 7:     The reward $r_t^k$ is revealed
 8:     The player updates its policy $\pi$
 9:     $t = t + 1$
10: **Until** t=T

---

# B    BACKGROUND

This section introduces some background concepts our approach builds upon, such as contextual bandit and Thompson Sampling.

**The contextual bandit problem**
Following (Langford & Zhang, 2008b), this problem is defined as follows. At each time point (iteration) $t \in \{1, ..., T\}$, an agent is presented with a *context (feature vector)* $\mathbf{x}_t \in \mathbf{R}^N$ before choosing an arm $k \in A = \{1, ..., K\}$. We will denote by $X = \{X_1, ..., X_N\}$ the set of features (variables) defining the context. Let $\mathbf{r}_t = (r_t^1, ..., r_t^K)$ denote a reward vector, where $r_t^k \in [0, 1]$ is a reward at time $t$ associated with the arm $k \in A$. Herein, we will primarily focus on the Bernoulli bandit with binary reward, i.e. $r_t^k \in \{0, 1\}$. Let $\pi : X \to A$ denote a policy. Also, $D_{c,r}$ denotes a joint distribution over $(\mathbf{x}, \mathbf{r})$. We will assume that the expected reward is a linear function of the context, i.e. $E[r_t^k | \mathbf{x}_t] = \mu_k^T \mathbf{x}_t$, where $\mu_k$ is an unknown weight vector (to be learned from the data) associated with the arm $k$.

**Contextual Thompson Sampling**
In this setting, we consider the general Thompson Sampling, where the reward $r_t^i$ for choosing arm $i$ at time $t$ follows a parametric likelihood function $Pr(r_t|\tilde{\mu}_i)$. Following (Agrawal & Goyal, 2013), the posterior distribution at time $t + 1$, $Pr(\tilde{\mu}_i|r_t) \propto Pr(r_t|\tilde{\mu}_i)Pr(\tilde{\mu}_i)$ is given by a multivariate Gaussian distribution $\mathcal{N}(\hat{\mu}_i(t+1), v^2 B_i(t+1)^{-1})$, where $B_i(t) = I_d + \sum_{\tau=1}^{t-1} x_\tau x_\tau^\top$, and where $d$ is the size of the context vectors $\mathbf{x}_i$, $v = R\sqrt{\frac{24}{\epsilon}dln(\frac{1}{\gamma})}$ with $R > 0$, $\epsilon \in ]0, 1]$, $\gamma \in ]0, 1]$ constants, and $\hat{\mu}_i(t) = B_i(t)^{-1}(\sum_{\tau=1}^{t-1} x_\tau r_\tau)$. At every step $t$, the algorithm generates a $d$-dimensional sample $\tilde{\mu}_i$ from $\mathcal{N}(\hat{\mu}_i(t), v^2 B_i(t)^{-1})$, for each arm, selects the arm $i$ that maximizes $x_t^\top \tilde{\mu}_i$, and obtains reward $r_t$.

---

**Algorithm 3** The Contextual Thompson Sampling Algorithm
---
1: **Initialize: for** $i = 1, ..., k$, $B_i = I_d$, $\hat{\mu}_i = 0_d$, $f_i = 0_d$.
2: **for** $t = 1, 2, ..., T$ **do**
3:     Receive context $\mathbf{x}_t$
4:     **for** $i = 1, ..., k$, sample $\tilde{\mu}_i$ from the $N(\hat{\mu}_i, v^2 B_i^{-1})$
5:     Choose arm $i_t = \arg \max\limits_{i \subset I} x(t)^\top \tilde{\mu}_i$
6:     Receive reward $r_k^i$
7:     $B_i = B_i + x_t x_t^T$, $f_i = f_i + x_t r_t^i$, $\hat{\mu}_i = B_i^{-1} f_i$
8: **end**

---

## C    EMPIRICAL EVALUATION

### C.1    DATASETS

We evaluated our approach on four imaging datasets: MNIST (LeCun, 1998), STL-10 (Coates & Ng, 2011), CIFAR-10 (Coates et al., 2011), Caltech-101 Silhouettes-28 (Griffin et al., 2007) and Warfarin (Consortium et al., 2009) (for details of each dataset, see Table 1). To simulate an online data stream, we draw samples from each dataset sequentially, starting from the beginning each time we draw the last sample. At each round, the algorithm receives reward 1 if the instance is classified correctly, and 0 otherwise. We compute the total number of classification errors as a performance metric.

*It is important to keep in mind that the bandit feedback (correct/incorrect classification) makes the classification problem significantly more challenging, as compared to the standard supervised learning, since the true label is never revealed in bandit setting unless the classification is correct. Thus, the classification accuracy in a bandit setting is expected to be lower than in the supervised learning setting.*

Table 1: Datasets

| Datasets | History | Instances | Features | Classes |
|---|---|---|---|---|
| MNIST | 10 000 | 20 000 | 784 | 10 |
| STL-10 | 20 000 | 10 000 | 1 000 | 10 |
| CIFAR-10 | 2 000 | 10 000 | 3 072 | 10 |
| Caltech-101 S | 671 | 8 000 | 784 | 101 |
| Warfarin | 528 | 5 000 | 93 | 3 |
| mix: MNIST/Warfarin | 10 528 | 10 000 | 93 | 13 |

We now describe some details of the experiments. For MNIST, we took 10,000 samples from the original test dataset (clearly, not using them later for testing) to pre-train the encodings, and 60,000 samples from the training dataset to simulate the online bandit with 10 arms corresponding to different digits. For STL-10, 100,000 samples of unlabeled data are used to pre-train the encodings; then the 5,000 test samples together with 8,000 training samples are combined to simulate the online bandit, again with 10 different arms corresponding to image classes[1]. For Caltech-101 Silhouettes-28 dataset, out of the original 8671 samples, 671 are used for pre-training and 8000 for online learning with 101 different arms (class labels). For CIFAR-10 dataset, 10,000 test set samples are used for pre-training, and 50,000 training samples are left for the online bandit with 10 arms (classes). For Warfarin dataset, 528 test set samples are used for pre-training, and 5,000 training samples are left for the online bandit with 3 arms (classes).

### C.2    NONSTATIONARY ENVIRONMENTS

We simulated several types of nonstationarity using the above datasets. As mentioned before, we assume that the input data arrive in batches, and the data distribution (i.e., the joint distribution of

---

[1]To speed up the computation, we squeezed input 27648-dimensional vectors into 1000-dimensional ones via linear stretching.

the context and reward) may change across those batches, while remaining stationary within each batch. We used the batch size of 1,000, and varied the number of embeddings $k$, using $k = 2, 4$, or 8, presenting average results over all $k$.

### C.2.1    NONSTATIONARY CONTEXT: VARYING CLUSTER DISTRIBUTION

To simulate changes in the context (input) distribution, we first clustered all samples in the corresponding pre-training data subset into $k$ clusters. Next, we generate a sequence of batches, where each batch contained a certain fraction of samples from different clusters, and these fractions were changing across the batches, i.e. the probability distribution of cluster membership was changing, simulating nonstationary input.

### C.2.2    NONSTATIONARY CONTEXT: NEGATIVE IMAGES

Another type of input nonstationarity involved introducing negative images as inputs with same semantics but different textures. Namely, with probability $p$, the negative image of the original image was presented as an input. Experiments were performed in two settings: *half* ($p = 0.5$) and *rand* ($0 < p < 1$ randomly assigned for each mini-batch), in both stationary and nonstationary context conditions, with both shuffled and unshuffled rewards (described later).

### C.2.3    NONSTATIONARY REWARD: MULTI-TASK ENVIRONMENT

Another type of nonstationarity was assuming that input samples may come from different domains (tasks), and thus can be associated with different subsets of labels (arms). For example, we combined 5,000 randomly selected training samples from each of the two selected domains, MNIST and Warfarin datasets, and extended the set of possible labels (arms) to include 10 labels from MNIST and 3 labels from Warfarin. We used linear stretching to make the input dimensions equal across the two domains. The algorithm had to assign a label to each input without any information about which domain the input came from.

### C.2.4    NONSTATIONARY REWARD: SHUFFLED CLASS LABELS

We further explored the multi-task setting by introducing a different type of nonstationary reward, where the class labels were shuffled, i.e. randomly permuted, in each batch.

## C.3    RESULTS

We explored different combination of the above nonstationarities. Table 2 summarizes our results for the nonstationary context due to varying cluster distribution, and for mixed-domain (multi-task) settings, with *unshuffled reward function*. As we can see, on three out of six datasets, baseline was still outperforming our embeddings. However, if we consider the mean accuracy in the entire set of experiments, the top three algorithms were: *universal embedding* (mean accuracy 28.83%), *baseline* (mean accuracy 27.78%), *mini-batch embedding* (mean accuracy 27.58%), respectively, suggesting the advantage of representation learning (embedding computation). Moreover, if we take a look at the whole iteration history, for example, for MNIST dataset (Figure 1), we observe that initially, the baseline CB (solid line) is considerably worse than embedding-based approaches, and requires a large number of iteration to finally catch up with them. Figures 2 and 3 show the history of reward accumulation for the STL-10 and CIFAR-10, demonstrating that the baseline is consistently outperformed by embedding selection methods.

Table 2: Nonstationary Environment with Unshuffled Labels

| Datasets | baseline | uE | mE | oE |
|---|---|---|---|---|
| MNIST | **37.24** | 34.44 | 29.00 | 22.32 |
| STL-10 | 10.29 | **15.81** | 14.77 | 13.43 |
| CIFAR-10 | 9.62 | **14.30** | 13.30 | 11.73 |
| Caltech-101 S | **1.18** | 1.14 | 1.09 | 1.06 |
| Warfarin | **62.58** | 56.70 | 56.10 | 56.92 |
| mix: MNIST/Warfarin | 45.76 | 50.58 | **51.21** | 47.74 |

Table 3: Nonstationary Environment with Shuffled Labels

| Datasets | baseline | uE | mE | oE |
|---|---|---|---|---|
| MNIST | 12.19 | **33.75** | 29.04 | 23.83 |
| STL-10 | 10.05 | **16.64** | 15.10 | 12.77 |
| CIFAR-10 | 10.23 | **14.83** | 13.13 | 11.60 |
| Caltech-101 S | 1.00 | 1.09 | 1.23 | **1.30** |
| Warfarin | 40.66 | **55.10** | 50.56 | 54.44 |
| mix: MNIST/Warfarin | 23.54 | 49.33 | **50.67** | 49.15 |

Table 4: Negative Environment with Unshuffled Labels

| Datasets | baseline | uE | mE | oE |
|---|---|---|---|---|
| MNIST half-stat | 13.50 | 14.70 | 14.02 | **16.18** |
| MNIST rand-stat | 13.72 | 17.14 | 15.53 | **17.70** |
| MNIST half-nonStat | 14.45 | 25.09 | 23.82 | **26.90** |
| MNIST rand-nonStat | 14.05 | 24.38 | 25.90 | **28.43** |
| STL-10 half-stat | 10.06 | **10.42** | 10.33 | 10.04 |
| STL-10 rand-stat | 9.77 | **12.34** | 12.33 | 10.41 |
| STL-10 half-nonStat | 9.88 | 10.99 | **12.29** | 11.56 |
| STL-10 rand-nonStat | 9.85 | 12.99 | **13.67** | 11.55 |
| Caltech-101 S half-stat | 0.98 | **10.04** | 7.98 | 6.94 |
| Caltech-101 S rand-stat | 0.94 | 10.93 | 8.40 | **11.68** |
| Caltech-101 S half-nonStat | 1.04 | 1.20 | **1.23** | 0.96 |
| Caltech-101 S rand-nonStat | 0.96 | 1.09 | **1.20** | 0.99 |

Next, Table 3 summarizes our results with *shuffled reward function*, for the nonstationary context due to varying cluster distribution, and for mixed-domain (multi-task) settings. Based on the mean accuracy in the entire experiment, the top three algorithms were: *universal embedding* (mean accuracy 28.46%), *mini-batch embedding* (mean accuracy 26.62%), *online embedding* (mean accuracy 25.52%), respectively. Furthermore, in this experiment, our embedding-based approaches always outperformed the baseline, suggesting that in a setting where reward functions are nonstationary, in addition to the nonstationary input environment, the advantage of representation learning is quite significant, as compared to standard CB (mean accuracy 16.28%). Note that, with nonstationary (shuffled) labels, the reward accumulated by the baseline CB remains significantly below the reward of embedding-based approaches, at all iterations (Figures 4-6). Thus, in a more challenging setting with both context and reward nonstationarities, the embedding-based approaches clearly outperform the standard contextual bandit.

Table 4 summarizes our results for the nonstationary online learning setting with *negative environments* and *unshuffled reward*. Based on the mean accuracy in the entire experiment, the top three

Table 5: Negative Environment with Shuffled Labels

| Datasets | baseline | uE | mE | oE |
|---|---|---|---|---|
| MNIST half-stat | 10.22 | 14.59 | 13.86 | **14.79** |
| MNIST rand-stat | 9.87 | **17.84** | 14.35 | 17.32 |
| MNIST half-nonStat | 10.78 | 23.02 | 22.33 | **26.84** |
| MNIST rand-nonStat | 11.27 | 27.34 | 24.87 | **28.36** |
| STL-10 half-stat | 9.66 | **11.51** | 10.73 | 10.60 |
| STL-10 rand-stat | 9.95 | 11.44 | **12.37** | 11.17 |
| STL-10 half-nonStat | 10.31 | 11.86 | **13.17** | 11.19 |
| STL-10 rand-nonStat | 9.2 | **12.62** | 12.49 | 11.59 |
| Caltech-101 S half-stat | 1.11 | **8.71** | 6.21 | 7.93 |
| Caltech-101 S rand-stat | 0.94 | **10.36** | 9.11 | 3.38 |
| Caltech-101 S half-nonStat | 1.06 | 1.03 | 1.00 | **1.29** |
| Caltech-101 S rand-nonStat | 1.08 | 1.05 | 1.13 | **1.16** |

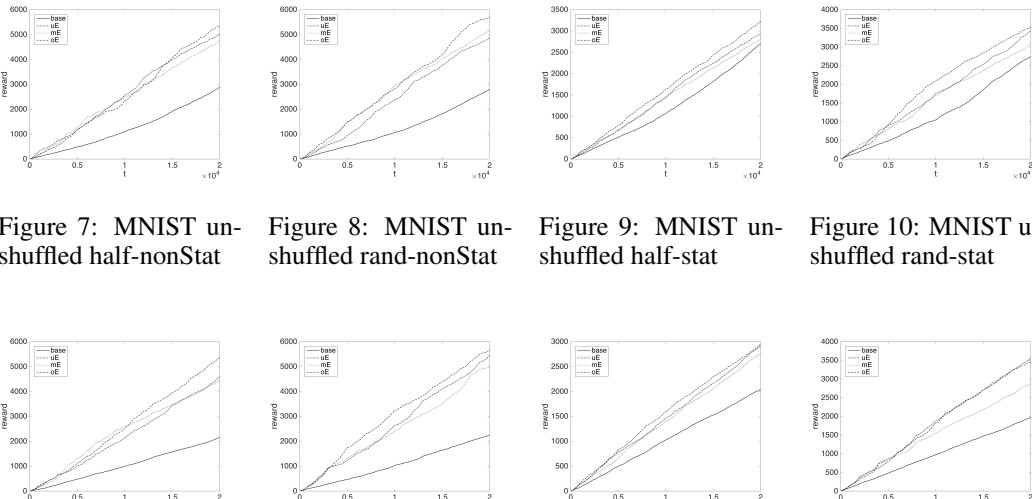

Figure 7: MNIST un-shuffled half-nonStat

Figure 8: MNIST un-shuffled rand-nonStat

Figure 9: MNIST un-shuffled half-stat

Figure 10: MNIST un-shuffled rand-stat

Figure 11: MNIST shuffled half-nonStat

Figure 12: MNIST shuffled rand-nonStat

Figure 13: MNIST shuffled half-stat

Figure 14: MNIST shuffled rand-stat

algorithms were: *online embedding* (mean accuracy 12.78%), *universal embedding* (mean accuracy 12.61%), *mini-batch embedding* (mean accuracy 12.23%), respectively. Again, the embedding-based approaches are always superior to the baseline CB; *online embedding* achieved the best performance among all methods on MNIST, while universal and batch embeddings were taking their turns outperforming the baseline on other datasets and settings.

Finally, Table 5 summarizes our results for the nonstationary online learning setting with negative environments and shuffled reward function. Based on the mean accuracy in the entire experiment, the top three algorithms were: *universal embedding* (mean accuracy 12.61%), *online embedding* (mean accuracy 12.14%), *mini-batch embedding* (mean accuracy 11.80%), respectively, further confirming the advantage of adaptive encoding over standard CB (mean accuracy 7.12%). In addition, the difference of textures under the same semantics introduced in this experiments demonstrated that embedding selection outperforms single universal embedding in most nonstationary cases.

Figures 7-14 visualize the details of reward accumulation over time by different methods, on MNIST data and all the settings from the Tables 4 and 5. The performance gap between the embedding-based approaches and the baseline is especially large in those settings. Furthermore, we can see that both adaptive, context-dependent embedding approaches (oE and mE) consistently ourperform the single-embedding approach (uE), with the *online embedding* emerging as the best one, especially with increasing number of iterations.

