# OpenReview forum: "Adaptive Representation Selection in Contextual Bandit with Unlabeled History"
_ICLR.cc/2018/Workshop — Reject_

### Official Review · AnonReviewer2 · 2018-03-05
**Where experiments on "semi-supervised" bandits confirms that combining clustering with embeddings is a strange idea**

**Rating:** 5
**Confidence:** 5

**Review:**

This experimental paper study a semi-supervised form of contextual bandits where an unlabeled history is used to build a good representation (an embedding) of the contexts before applying on-line predictions with "bandit feedback" (i.e. where feedback is only provided for the selected action/prediction).
The authors also propose a new algorithm called "ABaCoDe" which selects between different embeddings, each one computed on a different cluster of data. The clustering and the embeddings may be recomputed on-line with a mini-batch approach.

1- The experiments show that using unlabeled past data to build embeddings can improve contextual bandits;
2- The experiments also confirms that the (quite sophisticated) ABaCoDe algorithm is globally less efficient than than the (much simpler) "universal embedding" approach.

This is not a surprising result: the clustering procedure being a rough (lossy) form of embedding (a k-clustering maps each instance into a binary k-dimension space) it seems odd to combine it with (lossless) embeddings (a k-embedding maps each instance into a continuous k-dimension space). Using only the lossless "universal embedding" seems a much sound and  practical approach.


PRO:
- the experiments are honestly driven and show that semi-supervised learning works for contextual bandits
CON:
- the ABaCoDe approach seems pointless to me;
- the non-stationarity being a "non-assumption" is it difficult to obtain general conclusions: the experiments are only given for some specific forms of non-stationarity (described on page 3).

Minor remarks:
- The plots could be made more readable by giving the good prediction rates (#good predictions/ # steps) instead of the cumulative number of good predictions (reward);
- The batch semi-supervised performances (i.e. with full feedback) should be given as references

---

### Official Review · AnonReviewer1 · 2018-03-08
**Important problem but lacks crucials precisions**

**Rating:** 5
**Confidence:** 4

**Review:**

The papers states an interesting and ambitious problem but I fell that some points would merit some clarifications :
- The exact setting is not very clearly stated (what is the context, what is the reward -here I assume this is good classification of the presented label- , what are the parameters of the dynamics)
- The algorithm is not exactly new : some fixed clustering and one bandit by cluster (`Latent Bandits from O. Maillard even analyse a close scenario with a variable number of clusters).  The originality comes from the use of an embedding but the impact of having several embeddings is not so explicit. As an example the size of the embeddings are not provided (and not studied) while it may be important when building only one embedding to make it bigger to compete with the k others ones. In particular do the authors claims that learning several locals embeddings is possibly better than a big one?
- Plots are cumulated gains, while in bandits it is more common to plot the regret. Here it is hard to tell if the algorithm runned long enough (and the confidence interval are missing).
- Only reports scores for k=2 which make no clear sense when the datasets have 10 classes.
- Assuming  the task is classification it seems to performs poorly because the slope on cumulative gain on MNIST at the end of the plot is close to 0.5. If I'm correct this would mean that only half of the provided digits are correctly classified among a set of 5.

---

### Decision · Program_Chairs · 2018-03-20
**ICLR 2018 Workshop Acceptance Decision**

**Decision:**

Reject

**Comment:**

Based on the reviews, this paper has not been accepted for presentation at the ICLR workshop. However, the conversation and updates can continue to appear here on OpenReview.